

## New insights on the prevalence of drizzle in marine stratocumulus clouds based on a machine learning algorithm applied to radar Doppler spectra

Zeen Zhu[1], Pavlos Kollias[1,2], Edward Luke[2] and Fan Yang[2]

[1]School of Marine and Atmospheric Sciences, Stony Brook University, Stony Brook, NY, USA

[2] Environmental and Climate Sciences Dept, Brookhaven National Laboratory Upton, NY, USA

*Correspondence to:* Zeen Zhu (zeen.zhu@stonybrook.edu)

**Abstract**

The detection of the early growth of drizzle particles in marine stratocumulus clouds is important for studying the transition from cloud water to rainwater. Radar reflectivity is commonly used to detect drizzle; however, its utility is limited to larger drizzle particles. Alternatively, radar Doppler spectrum skewness has proven to be a more sensitive quantity for detection of drizzle embryos. Here, a machine-learning (ML) based technique that uses radar reflectivity and skewness for detecting small drizzle particles is presented. Aircraft in-situ measurements are used to develop and validate the ML algorithm. The drizzle detection algorithm is applied to three Atmospheric Radiation Measurement (ARM) observational campaigns to investigate the drizzle occurrence in marine boundary layer clouds. It is found that drizzle is far more ubiquitous than previously thought, the traditional radar reflectivity-based approach significantly underestimates the drizzle occurrence, especially in thin clouds with liquid water path lower than $50 \mathrm{~gm}^{-2}$. Furthermore, the drizzle occurrence in marine boundary layer clouds differs among three ARM campaigns, indicating that the drizzle formation which is controlled by the microphysical process is regime dependent. A complete understanding of the drizzle distribution climatology in marine stratocumulus clouds calls for more observational campaigns and continuing investigations.



# 1.Introduction

Clouds play an important role in the climate system and the accurate representation of their properties and feedbacks in Global Circulation Models (GCM) is essential for performing reliable future climate prediction (Cess et al., 1989;Bony et al., 2006;Vial et al., 2013). Among all the cloud types, marine stratocumulus is an important cloud type covering approximately 20% of the Earth's surface (Warren et al., 1986, 1988;Wood, 2012). Marine stratocumulus clouds significantly modulate the Earth's energy budget: on one hand, stratocumulus with high albedo strongly reflect incoming solar radiation back to space; on the other hand, as stratocumulus clouds have similar temperature with surface, they emit comparable amount of longwave radiation as the surface and do not significantly affect the infrared radiation emitted to space. Thus, overall the stratocumulus have a strong cooling effect to the climate system. (Hartmann et al., 1992). It is estimated that only a small increase of the marine stratocumulus coverage can compensate for the increased temperature induced by the greenhouse gas effect (Randall et al., 1984). Despite the considerable influence on the climate, large uncertainties persist in the representation of marine stratocumulus in GCMs due to a lack of understanding of the cloud properties and the associated processes. (Stephens, 2005;Klein et al., 2017) One important issue is the underrepresentation of the transition from cloud water to rainwater, i.e. the autoconversion process. (Stephens et al., 2010;Michibata and Takemura, 2015). (Paluch and Lenschow, 1991;Yamaguchi et al., 2017). A misrepresentation of the autoconversion process in GCM's can affect not only the hydrological cycle but also generate compensating errors in the aerosol-cloud interactions (Michibata and Suzuki, 2020).

The core component of autoconversion is the production and growth mechanisms of drizzle drops. Drizzle, by definition, refers to liquid droplets with a diameter between 40 μm and 500 μm (Wood, 2005a;Glienke et al., 2017;Zhang et al., 2021). Drizzle is frequently observed in the warm cloud system and can modulate the cloud organizational structure and the boundary layer system in several ways: the drizzle production process tends to warm the cloud layer and stabilize the boundary layer, which reduces cloud top entrainment and produces thicker clouds (Wood, 2012;Nicholls, 1984;Ackerman et al., 2009); the coalescence process can reduce cloud droplet concentration and cause cloud precipitation (Wood, 2006); furthermore, drizzle also plays a critical





role in the formation of the open-cell pattern of stratocumulus (Wang and Feingold, 2009;Feingold
et al., 2010) and tends to promote the stratocumulus to cumulus transitions process (Paluch and
Lenschow, 1991;Yamaguchi et al., 2017).

Despite the importance role of drizzle plays on the marine boundary layer, a thorough
understanding of its existence is incomplete due to the detection limitation. Historically, in-situ
and remote sensing measurements have been used to detect drizzle in cloud (Leon et al.,
2008;Wood, 2005a;Wu et al., 2015;Yang et al., 2018;VanZanten et al., 2005). In-situ
microphysical probes can provide size-resolved microphysical properties, importantly, Drop Size
Distribution (DSD), from which drizzle drops can be easily identified according to their definition.
The disadvantage of in-situ observations is the limited datasets collected during field campaigns,
making it challenging to provide long term statistical analyses. Millimeter-wavelength radar,
commonly known as cloud radar, is widely used for cloud/drizzle detections (Kollias et al., 2007a).
The total received backscatter power of droplets is converted to radar reflectivity factor, which is
independent of the radar wavelength in the cloud/drizzle regime, and is proportional to the sixth
power of the diameter of the particles in the radar resolution volume[1]. Compared with cloud
droplets, drizzle drops have larger diameters, which produce greater reflectivity, and this signature
is widely used to differentiate cloud/drizzle signals. Different reflectivity thresholds, ranging from
-15dBZ to -20dBZ, have been applied in previous studies to identify drizzle existence (Frisch et
al., 1995;Liu et al., 2008;Comstock et al., 2004). Nevertheless, this reflectivity-based technique
has obvious drawbacks. As reflectivity is the summation of the backscattered power from all the
droplets in a radar volume, the reflectivity threshold can detect the presence of drizzle drops only
when their contribution to the total radar backscatter exceeds that of the cloud droplets. More
specifically, when cloud droplets dominate the reflectivity signal, even if drizzle drops exist, they
fail to be detected as the total reflectivity is usually lower than -20 dBZ; this indicates that the
reflectivity-based technique is unable to detect small drizzle particles (Kollias et al., 2011b).

Besides reflectivity, another radar observed quantity which is sensitive to the presence of drizzle
is the skewness of the radar Doppler spectrum (hereafter skewness). Skewness is the third moment
of the radar-observed Doppler spectrum and is a measure of the asymmetry of the spectrum. For

---

[1] It is noted that attenuation is not considered in this study.



cloud droplets, Doppler spectra are on average symmetric with skewness equal to zero; as drizzle
drops grow and start falling, their terminal velocity is recorded in the fast-falling part of the
Doppler spectra, which has greater backscatter power than the power contributed by cloud droplets,
leading to  asymmetric spectra with a non-zero skewness (Kollias et al., 2011b;Luke and Kollias,
2013). The capability of using skewness to detect early drizzle development stages was
demonstrated in Acquistapace et al. (2019). In Acquistapace et al. (2019) a threshold of the
skewness is used as part of the detection algorithm. Considering the noisiness in the estimation of
the third moment of the radar Doppler spectrum, the use of a fixed threshold value can lead to
considerable misclassifications.  Here, a supervised Machine Learning (ML) algorithm is used to
provide a more robust detection of drizzle particles in warm stratiform clouds. First, in-situ DSD
measurements are used as input to a sophisticated radar Doppler spectrum simulator that can
accurately represent the performance of the ARM profiling cloud radars in estimating the
corresponding radar-observed reflectivity and skewness. Next, the ML algorithm is trained from
2 months of in-situ observations to generate a classification model; the classification results from
one case study will be presented and compared against the in-situ measurements. Finally,
comprehensive datasets from three ARM observational campaigns are used to investigate drizzle
occurrence and demonstrate the omnipresence of drizzle in marine stratocumulus clouds.

**2.Instruments and Data**

The data used in this study are collected from three observatories operated by the U.S. Department
of Energy's Atmospheric Radiation Measurement (ARM) facility. The Eastern North Atlantic
(ENA) is a permanent observational site established on Graciosa Island in the Azores archipelago
in 2013 as representative of a maritime environment. The Aerosol and Cloud Experiments in the
Eastern North Atlantic (ACE-ENA) field campaign was conducted in the vicinity of the ENA site
from June 2017 to February 2018. The Gulfstream-1 aircraft was deployed during ACE-ENA to
provide in-situ measurements. The Marine ARM GPCI Investigation of Clouds (MAGIC)
campaign was based on a mobile observatory facility traversing between Los Angeles, California,
and Honolulu, Hawaii, from October 2012 to September 2013. Measurements of Aerosols,
Radiation, and Clouds over the Southern Ocean (MARCUS) was a field campaign conducted from
October 2017 to April 2018 along the route between Hobart, Australia, and the Antarctic. All of



the observational campaigns were equipped with a variety of instruments which provide
comprehensive datasets being used in this study.

The primary instrument being used in this study is the cloud radar: a Ka-Band ARM Zenith Radar
(KAZR) was operated at ENA and MAGIC and a W-Band ARM Cloud Radar (WACR) was used
during MARCUS. The KAZR and WACR are both vertically pointing with 30 m range resolution;
the temporal resolution of the WACR and KAZR used at ENA is 2 s, while the temporal resolution
of the KAZR used for MAGIC is 0.36 s. To make the observations comparable, radar moments
from MAGIC are averaged over 2 s to be consistent with the ones collected at ENA and MARCUS.
Radar reflectivity and Doppler skewness are obtained from the Microscale Active Remote Sensing
of Clouds (MicroARSCL) product (Kollias et al., 2007b). Radar reflectivity at ENA and MAGIC
is calibrated with surface-based measurements of the raindrop PSD using a disdrometer (Gage et
al., 2000;Kollias et al., 2019). At MARCUS, a disdrometer is not suitable for radar calibration thus
instead we follow Mace et al. (2021) by adding 4.5 dB to the reflectivity for WACR calibration.
In addition, a ceilometer and microwave radiometer (MWR) are used to estimate cloud base height
and liquid water path (LWP). The time resolution of the MWR and ceilometer are 10 s and 15 s
respectively. Besides the surface-based observations, in-situ measurements from ACE-ENA
during the intensive observation period 1 (IOP1) which was conducted from 21 June to 20 July in
2017 are also used in this study. The DSD of hydrometeors with diameter ranging from 1.5 μm to
9075 μm are characterized using combined measurements from the fast cloud droplet probe
(FCDP), 2-dimensional stereo probe (2D-S) and high-volume precipitation spectrometer (HVPS-
3). Liquid water content is measured using a multi-element water content system and a Gerber
probe.

**3.Methodology**

As Doppler skewness is a sensitive indicator of weak drizzle signals, the focus of the methodology
is to synthesize this quantity with reflectivity to construct a robust drizzle detection algorithm.
Thus, the key issue lies in the challenging task of determining the appropriate reflectivity/skewness
combination to identify drizzle signals. Here we address this problem in a novel way: first we
identify the existence of cloud/drizzle based on in-situ observed DSDs; then a well-established



Doppler spectrum simulator is applied to emulate the radar observed spectrum for the given DSD
and estimate the corresponding reflectivity and skewness. Finally, the resulting collection of well-
defined cloud/drizzle datasets is trained by a machine learning algorithm to resolve the drizzle
identification function.

**3.1 Doppler spectrum simulation**

According to previous studies, liquid droplets with diameter exceeding 40 μm are defined to be
drizzle (Wood, 2005a;Zhang et al., 2021). We follow this definition to classify the in-situ observed
DSD: cloud/drizzle are defined by the maximum diameter in the DSD being smaller/larger than
40 μm. Example DSDs of cloud-only and mixed cloud-drizzle conditions are shown in Fig. 1a and
Fig. 1c. Next, the Doppler spectrum simulator developed by Kollias et al. (2011a) is applied to
generate the radar-observed Doppler spectrum based on the in-situ DSD. The associated simulator
parameters are set as follows: Doppler spectra are generated with 256 FFT bins and a Nyquist
velocity of $\pm6$ m/s, which correspond to the KAZR configuration operated by ARM (Kollias et
al., 2016); turbulence broadening ($\sigma_t$) is set as 0.2m/s which is obtained from local observations:
for radar observation with reflectivity smaller than -20 dBZ, Doppler spectra width is mainly
contributed by turbulence and can be used to estimate $\sigma_t$. The KAZR-observed spectral width
collected from the ACE-ENA IOP1 indicate that the mean value of the $\sigma_t$ is estimated as 0.2 m/s
(Fig. S1). Finally, radar noise is simulated by adding random perturbation to the Doppler spectra;
positive velocity indicates downward motion. A detailed description of the Doppler spectrum
simulator application is found in Zhu et al. (2021). Once a spectrum is generated, a post-processing
algorithm (Kollias et al., 2007b) is used to eliminate noise (Hildebrand and Sekhon, 1974) and to
estimate the Doppler moments, i.e. reflectivity and skewness. To demonstrate that the simulator
can represent radar observations, the simulated reflectivity and skewness are compared with
KAZR observations (Fig. S2) and shows consistent ranges and distribution pattern, indicating that
the simulated radar moments are capable to represent the real observation signal. The relatively
large fraction of the in-situ measurements with dBZ > -20 in Fig. S2 is likely caused by the
different observational strategies between in-situ and KAZR measurements (Wang et al., 2016).



Fig. 1b and 1d show examples of the simulated Doppler spectra along with the estimated
reflectivity and skewness for a cloud-only and mixed cloud-drizzle DSD. It is noticed for the
drizzle case (Fig. 1d), reflectivity is well below the conventional threshold (-20 ~ -15 dBZ) used
for drizzle detection and is unable to discriminate it from the cloud-only case (Fig. 1b). Skewness,
however, shows a significant difference between drizzle (0.5) and cloud (0), emphasizing the
importance of including skewness as an additional indicator for drizzle detection.

**3.2 Machine Learning algorithm application**

From the IOP1 of ACE-ENA, 6000 in-situ observed DSDs (2000 for cloud-only and 4000 for
mixed cloud-drizzle) are selected from the cloudy samples defined as having liquid water content
larger than 0.01 gm$^{-3}$ (Zhang et al., 2021). For each DSD, the spectrum simulator is applied to
estimate the reflectivity and Doppler skewness. The distribution of these two quantities for all the
selected datasets is shown in Fig. 2. It shows that drizzle with positive skewness tends to be
associated with reflectivity lower than -20 dBZ. For reflectivity ranging from -35 to -25 dBZ and
skewness around zero, the drizzle signal overlaps with cloud; this region represents the early stage
of drizzle initiation with low reflectivity and indistinguishable skewness features compared with
cloud signals.

In order to determine the classification boundary to distinguish cloud/drizzle categories (i.e.
red/blue points in Fig. 2), we apply a supervised machine learning algorithm which is widely used
in classification-related problems, the Support Vector Machine (SVM) (Cortes and Vapnik,
1995;Vapnik et al., 1997). SVM handles complicated data classification tasks by solving
optimization relationships and finding the optimal classification equations in the feature space.
There are three reasons to use SVM in this study: 1) SVM is nonparametric and thus does not
require specification or assumption of the classification equation; 2) By applying the appropriate
kernel, SVM can generate a non-linear classification boundary to classify non-linearly separable
datasets; 3) The decision boundary resolved by SVM will separate the categories with maximum
distance; this is a distinctive feature of the SVM algorithm which is extensively used in a variety
of areas (Ma and Guo, 2014).



For the collected cloud/drizzle datasets, 80% of them are used for training, and the remain 20%
for validation. Inputs to the SVM are Doppler skewness and reflectivity, where the reflectivity
from -50 dBZ to 0 dBZ is normalized from -1 to 0; the output is classified as either cloud (0) or
drizzle (1). Here the Radial Basis Function (RBF) with two tuning parameters, $\Gamma$ and C, is used as
the SVM kernel (Keerthi and Lin, 2003). The RBF kernel is one of the most widely used kernels
due to its similarity to the Gaussian distribution. The $\Gamma$ parameter determines the curvature of the
decision boundary with a high value indicating more curvature for capturing the complexity of the
dataset; C is a regularization parameter to set the classification accuracy versus the maximization
of the decision function margin; a lower C leads to a larger margin, and a simpler decision function
at the cost of training accuracy. Following Davis and Goadrich (2006), we use precision/recall to
evaluate the performance of the classification outcome. In this study, precision refers to the number
of correct drizzle detections divided by total drizzle detections reported by the SVM, and recall
refers to the number of the correct drizzle detections divided by the number of true drizzle
occurrences in dataset. Different combinations of RBF parameters with $\Gamma$ ranging from 1 to 500
and C from 1 to 1000 are applied, with the classification outcome shown in Table 1. Here we
choose $\Gamma = 50$ and $C = 1$ as the preferred parameters to produce classification results with precision
and recall as 98% and 85%, respectively. That is, for the cloud-drizzle dataset collected at ACE-
ENA, at most, 85% of the drizzle can be detected by the algorithm and among the detection
outcomes, 98% are true drizzle signals.

The resolved classification boundary is shown as the black line in Fig. 2. We can see the algorithm
reasonably separates the cloud/drizzle clusters; the resolved skewness threshold being used to
distinguish cloud/drizzle is around $\pm 0.2$, and the maximum reflectivity used for classification is -
20dBZ. These values are consistent with previous studies (Frisch et al., 1995;Liu et al.,
2008;Kollias et al., 2011b;Acquistapace et al., 2019).  We further estimate the cumulative
distribution function (CDF) of the correctly detected drizzle samples as a function of dBZ from
the ML technique (magenta solid line in Fig. 2) and from the traditional method with reflectivity
threshold ranging from -20 to -15 dBZ. (magenta shading in Fig. 2). It is noticed that drizzle can
be detected with dBZ <-30 from the ML method; this value is significantly lower than for
traditional thresholds in use. The ML method is more sensitive to the weak drizzle signals than the
dBZ thresholds that have been proposed. Specifically, compared to the ML technique, 35% and



21% of the drizzle are missed by the reflectivity threshold approach when using dBZ >-20 and
dBZ >-15, respectively. Another important implication of this result is that dBZ >-15 is
traditionally applied by CloudSat to identify light rain incidence (Haynes et al., 2009); here we
demonstrate that a more robust threshold is likely to be much lower.

Besides the encouraging performance of the ML technique, some noticeable issues can be
identified: 1) Compared with the true CDF of the drizzle fraction (dotted magenta line in Fig. 2),
20% of drizzle is undetected. This missing drizzle subset, as explained previously by the
overlapping area, is composed of tiny drizzle embryos that have yet to develop distinctive features
compared with their cloud counterparts. 2) Another issue is the unrealistic broadening of the
classification boundary for reflectivity lower than -35dBZ; this issue is related to the kernel being
applied in the SVM algorithm. Since drizzle rarely exists below -35 dBZ, this issue will not affect
the classification performance as far as we are concerned.

**4.Results**

The ML-based drizzle detection algorithm is applied to the dataset collected at three ARM
observatories. First, an example case is presented for which aircraft observations are available and
the corresponding in-situ measurements are used to demonstrate the performance of the algorithm.
Then, the drizzle occurrence on classified stratocumulus clouds at ENA, MARCUS and MAGIC
observatories are presented; the differences of the drizzle occurrence from the proposed machine
learning based algorithm (hereafter MLA) and the traditional dBZ-based algorithm (hereafter
dBZA) are compared to indicate that drizzle occurrence in stratocumulus clouds is far more
frequent that has been previously suggested. For the dBZA, we use reflectivity >-17 dBZ for
drizzle identification, while the application of other thresholds ranging from -20 to -15 dBZ did
not affect the results as discussed.

**4.1 Single cloud layer case**

For the selected case (Fig. 3), a thin cloud layer with thickness around 150m is identified. Cloud
signals is very weak with 99% of reflectivity lower than -17 dBZ. However, considerable large



skewness values shown in Fig. 3b indicates the presence of the drizzle particles. The classification
results from the MLA classification are shown in Fig. 3c, it can be seen that drizzle is omnipresent
and spread throughout the cloud layer, mixed with cloud-only detections.

Here the in-situ observed DSD is used to verify the MLA detection. On June 30th, 2017, aircraft
measurements were conducted from 09:27 to 13:16 UTC. We constrained the in-situ
measurements to be within 20 km of the ENA observatory (Fig. 4). Considering that the average
in-cloud wind speed is 3.7 m/s, the distance of 20 km is equivalent to around 1.5 hour of KAZR
observations; thus, the radar measurements from 08:00 to 13:30 UTC are selected to match the
aircraft observations. We assume the signal of the drizzle/cloud occurrence collected from the in-
situ measurements can be used to verify the drizzle presence observed from KAZR. For the
selected period, drizzle occurrence is 47% from the MLA detections and 65% from the in-situ
observations. The 18% of the missing drizzle by MLA is largely attributed to the "overlapping
area" shown in Fig. 2 indicating the early stage of drizzle embryos which are indistinguishable
from cloud droplets. Nevertheless, this comparison provides strong evidence that drizzle is widely
present in the cloud layer for the selected case and demonstrates that the classification results from
MLA are reliable. Contrastingly, negligible drizzle signals (0.05%) are detected with the
reflectivity-based (dBZ >-17) technique.

**4.2 Drizzle occurrence at ARM campaigns**

During the operational periods of ACE-ENA, MARCUS and MAGIC, single-layer marine
stratocumulus clouds are selected with cloud top temperature greater than 0 ℃ and cloud top
height lower than 4000 m. The moving standard deviation of cloud top height within 30-minutes
($\sigma$) is calculated and profiles with $\sigma$ larger than 200 m are excluded to reject non-stratocumulus-
type clouds. LWP retrievals are biased when MWR is wet; thus, radar profiles with their lowest
range gates containing hydrometeor detections are considered to be precipitation and are removed
from the analysis. A complete list of the days being used is shown in Table 2. In total, 204, 72, and
215 hours of radar observation were selected from the ACE-ENA, MARCUS and MAGIC
campaigns.





In order to composite cloud layers with different thickness, cloud height is normalized between 0
to 1 as:
$$h = \frac{H - H_b}{H_t - H_b}$$


Where H is the physical height of a given radar gate, $H_t$ and $H_b$ is the cloud top and base height.
h=0 represents cloud base and h=1 indicates cloud top.

Drizzle occurrence is calculated as the number of drizzle detections divided by the total observed
signals in each normalized height bin (0.1) and LWP bin (50 g m$^{-2}$). The drizzle occurrence being
detected from both methods at the three ARM observatories are shown in Fig. 5. For all the
observational site/campaigns, drizzle is more likely to occur as LWP increases. This tendency
holds true despite the drizzle detection method being used. However, for each observational
campaign, drizzle occurrence detected from MLA (Fig. 5 a, b, c) is always larger than from dBZA
(Fig. 5 d, e, f). This difference becomes significant especially for thin clouds with low LWP: when
LWP is under 50 g m$^{-2}$, or equivalently, cloud thickness is less than 200 m (Fig. 6), drizzle
occurrence being detected from dBZA is around 0.1 while it is 0.4~0.5 from MLA.  This result
clearly indicates that the traditional drizzle detection method based on a reflectivity threshold
significantly underestimates the true drizzle occurrence, especially in thin cloud layers. To
quantitatively describe the detection performance, we estimate the relative percentage difference
of the drizzle detections between two methods as follows:
$$P_{LWP} \ (\%) = \frac{N_{MLA,LWP} - N_{dBZA,LWP}}{N_{MLA,LWP}} * 100$$

Where $N_{MLA,LWP}$ and $N_{dBZA,LWP}$ indicate the number of the drizzle detection by MLA and dBZA
respectively for a given LWP category. The results (Fig. 7a) indicate that when LWP is smaller
than 50 g m$^{-2}$, which frequently occurs under the ENA and MAGIC campaigns (Fig. 7b), 90% of
drizzle are missed by dBZA at ENA and MARCUS, and 60 % of drizzle is undetected at MAGIC
compared with MLA. An application of a relative lower reflectivity threshold with dBZ< -20, to
some degree, mitigate the missing drizzle detections compared with MBL, but still with   50~80%
of the drizzle being undetected (shading area in Fig. 7a).





Besides the considerable drizzle signals missed by dBZA, another implication to be noted is the
difference of drizzle distribution among the three ARM campaigns. Specifically, large drizzle
fractions tend to occur in the upper part of cloud at ENA and in the lower parts of cloud at
MARCUS and MAGIC (Fig. 5). When compared with MLA, the missing drizzle detections based
on dBZA are much more significant for ENA/MARCUS than for MAGIC (Fig. 7a). The different
drizzle distribution pattern suggests that clouds among these three campaigns might have different
microphysical properties and processes that controls the drizzle initiation. For instance, the
contrasting thermodynamics environment among the ARM campaigns with low/high temperature
and humidity at MARCUS/MAGIC might leads to different autoconversion process which control
the drizzle formation. In particular, we suspect that a more humid environment under MAGIC will
benefits the generation of larger cloud droplets compared with the other campaigns (Laird et al.,
2000;Zhou et al., 2015). Fig. 8 supports this hypothesis by showing that the mean cloud reflectivity
at MAGIC is 8dB larger than it is at the other two campaigns for LWP smaller than 100 $\text{gm}^{-2}$. The
relatively large dBZ for small LWP, to some degree, mitigates the underrepresented drizzle
detection by the reflectivity-based method.

**5.Conclusion and Discussion**

Building on the concept that radar Doppler spectra skewness is more sensitive to drizzle presence,
a new method of detecting drizzle in marine boundary clouds is presented. In-situ observed DSDs
are used to unambiguously classify cloud and drizzle particles; then, a radar Doppler spectra
simulator is applied to estimate the expected radar-observed reflectivity and skewness. Extensive
datasets collected from the ACE-ENA campaign are trained via the ML-based algorithm to
optimally determine a classification equation of cloud/drizzle. The proposed algorithm is validated
by the in-situ measurements to successfully detect weak drizzle signals, which are completely
missed by the traditional reflectivity-based technique.

The drizzle/cloud classification outcome of a thin cloud layer observed on June 30, 2017 at ENA
was presented to show the performance of the detection algorithm.  It was found that even for thin
cloud with thickness less than 150 m, a significant amount of drizzle already exists; this
classification result is further verified by the in-situ observations. Furthermore, a statistical





369 analysis compares the drizzle occurrence from two detection methods at the ACE-ENA, MARCUS

370 and MAGIC field campaigns. The results indicate that drizzle is ubiquitous in cloud layers and its

371 existence has been significantly underestimated by conventional reflectivity-based methods,

372 especially in thin cloud layers. The drizzle occurrence and vertical structure differ among the three

373 campaigns, indicating that drizzle formation and distribution in marine stratocumulus clouds might

374 be regime dependent, determined by microphysical and dynamical process in the local region. In

375 this study, data from the three observational campaigns are used to explore the drizzle frequency

376 of marine stratocumulus in middle/high latitude regions; however, it is quite possible that the

377 drizzle occurrence from other locations might differ from the presented results. A complete

378 understanding of the drizzle climatology in marine stratocumulus clouds calls for more campaign

379 observations and continuing investigations.

380

381 The results in this study provide a new perspective for viewing drizzle existence in radar

382 observations with the hope of shedding light on several critical topics in the warm cloud studies:

383 1) In most microphysics retrieval algorithms, the existence of drizzle particles is determined by a

384 reflectivity threshold. However, this study shows the presence of significant drizzle drops during

385 low reflectivity conditions (lower than -20 dBZ) and a lack of considering this may lead to a certain

386 degree of the retrieval uncertainty; 2) Drizzle production mechanisms are widely regarded as a

387 critical missing piece of the puzzle in warm cloud research (Takahashi et al., 2017). Particularly,

388 the parameterization schemes of the autoconversion/accretion processes in numerical models have

389 large variations among each other, leading to significant uncertainty in future climate predictions

390 (Michibata and Suzuki, 2020;Wood, 2005b). The results presented in this study can be used to

391 verify the proposed parameterization schemes by comparing the drizzle climatology. 3)

392 Furthermore, the novel utilization of in-situ and remote sensing synthesis of observations presented

393 in this study yields insights on the potential of combined multi-platform observations to investigate

394 the microphysical processes in warm clouds.








***Data availability:*** The ARM observational datasets are available at the ARM Data Center. The
KAZR data (kazrge) can be accessed via http://dx.doi.org/10.5439/1025214. The ceilometer
dataset (ceil) can be accessed via http://dx.doi.org/10.5439/1181954. The retrieved LWP product
(mwrret2turn) can be accessed via http://dx.doi.org/10.5439/1566156. The in-situ observation
during the ACE-ENA campaign can be accessed via
https://adc.arm.gov/discovery/#/results/iopShortName::aaf2017ace-ena.
***Supplement***: The supplement related to this article is available online at:
***Author contributions***: Z.Z. designed the methodology and performed the analysis. P.K.
contributed the design of the study. E.L provide the MicroARSCL datasets. F.Y. assisted in the
interpretation of results. Z.Z. prepared the manuscript with contributions from all co-authors.
***Competing interests:*** The authors declare that they have no conflict of interest.

***Acknowledgments***: We would also like to acknowledge the data support provided by the
Atmospheric Radiation Measurement (ARM) Program sponsored by the U.S. Department of
Energy.
***Financial support:*** Z. Z.'s contributions were supported by the U.S. Department of Energy (DOE)
ASR ENA Site Science award. P. K., E. L. and F. Y. were supported by the US Department of
Energy (DOE) Atmospheric System Research Program under contract DE-SC0012704.



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

Figures and Tables:

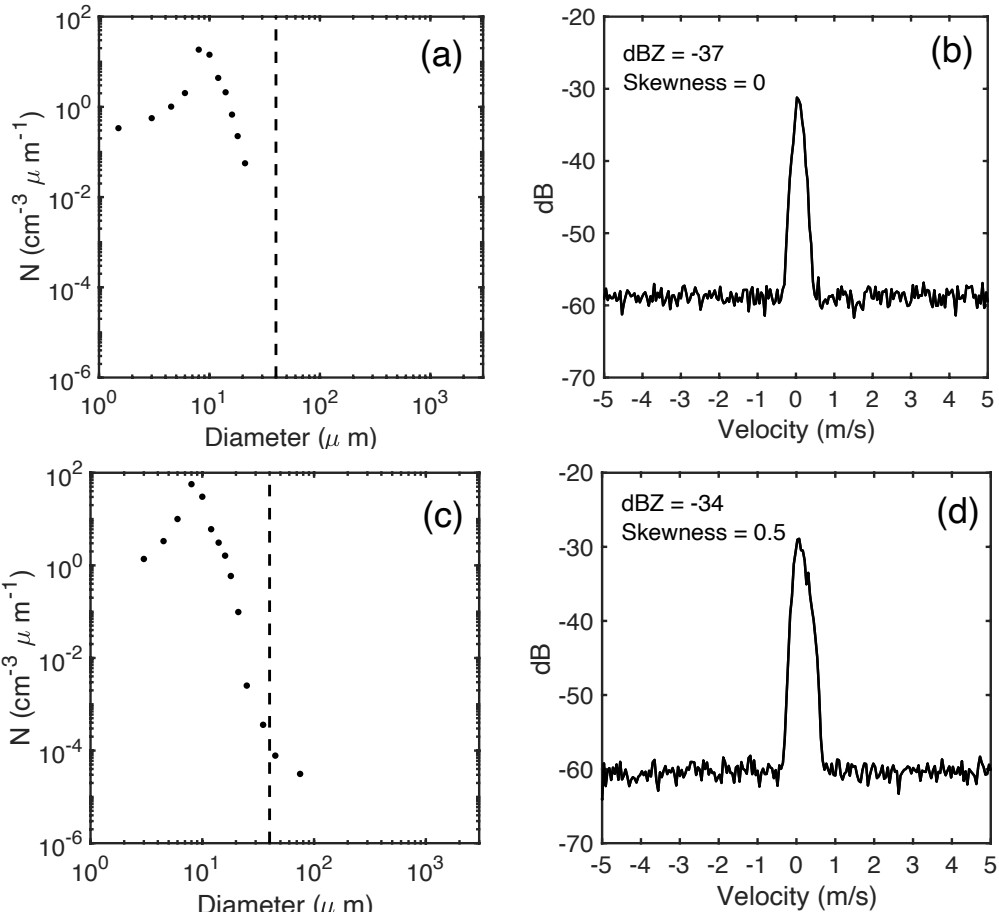


Figure 1: In-situ observed DSD of cloud-only (a) and the corresponding simulated Doppler Radar
spectrum (b), reflectivity and skewness of the spectrum are indicated in the upper left corner.  (c)
and (d) are same as (a), (b) but for mixed cloud-drizzle DSD.  The dash line in (a), (c) indicates
diameter with 40 $\mu m$.













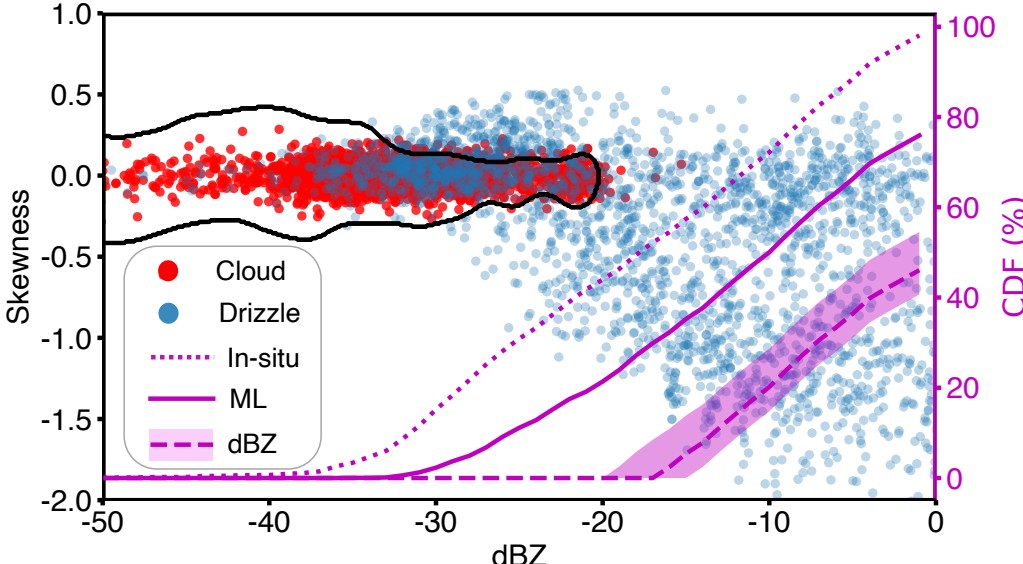

Figure 2: Distribution of the cloud-only (red points) and mixed cloud-drizzle (blue points) samples
from the in-situ observation over the reflectivity-skewness space. The black line indicates the
classification boundary of cloud/drizzle resolved by Machine Learning algorithm. Right axis
indicates the CDF of all correctly identified drizzly samples as a function of reflectivity by each
method: dotted magenta line is for the in-situ observations, which represents the true value; solid
magenta line is for the ML technique; the magenta shading is for the reflectivity-based technique
with upper boundary using dBZ > -20 and lower boundary using dBZ > -15; the dashed magenta
line is for the reflectivity-threshold technique with dBZ > -17.














Table 1: Precision(P) and Recall(R) of the drizzle/cloud classification outcome for different
combination of C and Γ. The dark shaded cell represents the classification performance for the
selected parameters (C=1, Γ=50) being used in the study.

| Γ $\diagdown$ C | 1 | 10 | 50 | 100 | 200 | 500 |
|---|---|---|---|---|---|---|
| 1 | 0.99(P) 0.82(R) | 0.98(P) 0.85(R) | 0.98(P) 0.85(R) | 0.98(P) 0.85(R) | 0.97(P) 0.86(R) | 0.92(P) 0.87(R) |
| 10 | 0.99(P) 0.84(R) | 0.98(P) 0.85(R) | 0.98(P) 0.85(R) | 0.98(P) 0.85(R) | 0.94(P) 0.85(R) | 0.91(P) 0.86(R) |
| 50 | 0.99(P) 0.84(R) | 0.98(P) 0.85(R) | 0.98(P) 0.85(R) | 0.97(P) 0.85(R) | 0.93(P) 0.86(R) | 0.89(P) 0.87(R) |
| 100 | 0.99(P) 0.84(R) | 0.98(P) 0.85(R) | 0.98(P) 0.84(R) | 0.96(P) 0.85(R) | 0.92(P) 0.86(R) | 0.89(P) 0.87(R) |
| 200 | 0.98(P) 0.85(R) | 0.98(P) 0.84(R) | 0.98(P) 0.84(R) | 0.95(P) 0.85(R) | 0.91(P) 0.86(R) | 0.89(P) 0.87(R) |
| 500 | 0.98(P) 0.85(R) | 0.98(P) 0.84(R) | 0.98(P) 0.85(R) | 0.94(P) 0.86(R) | 0.91(P) 0.86(R) | 0.88(P) 0.87(R) |
| 1000 | 0.98(P) 0.85(R) | 0.98(P) 0.84(R) | 0.97(P) 0.84(R) | 0.94(P) 0.86(R) | 0.90(P) 0.86(R) | 0.88(P) 0.88(R) |





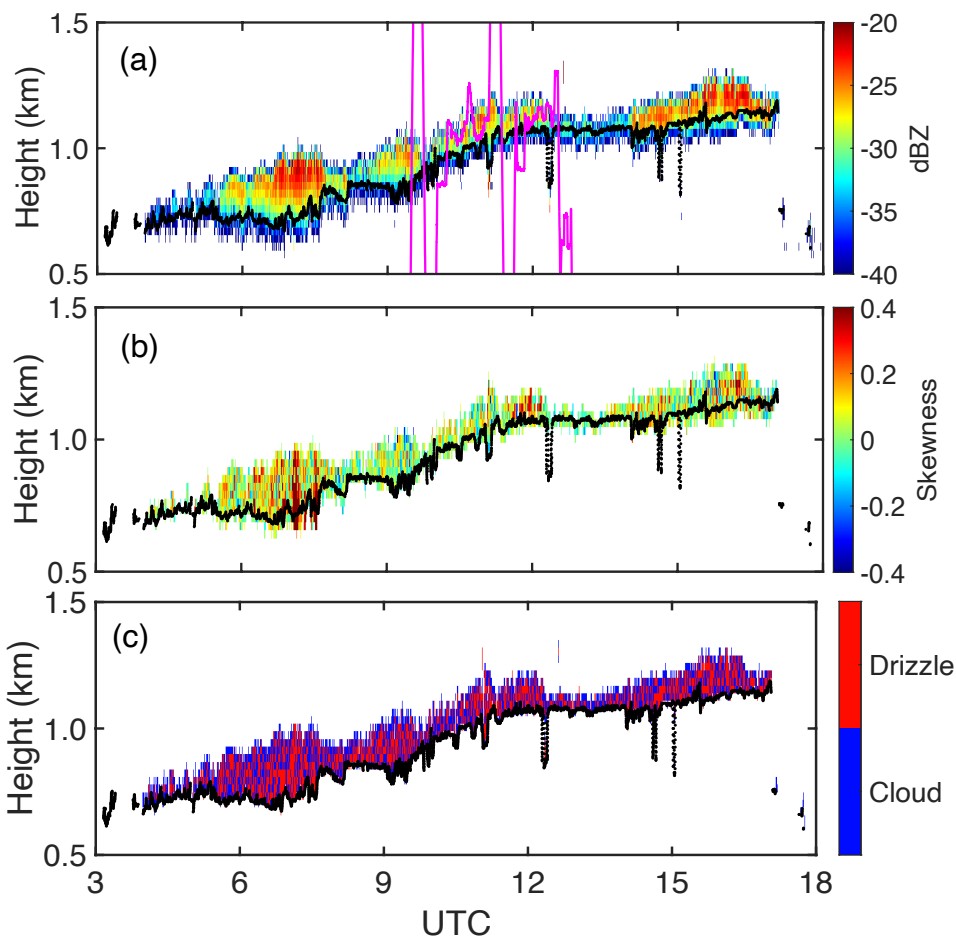

Figure 3: Reflectivity (a), skewness (b) and the classification mask (c) on June 30, 2017, at ENA
site. Black line indicates the ceilometer-determined cloud base, magenta line in (a) indicates
altitude track of the aircraft during the observation period.






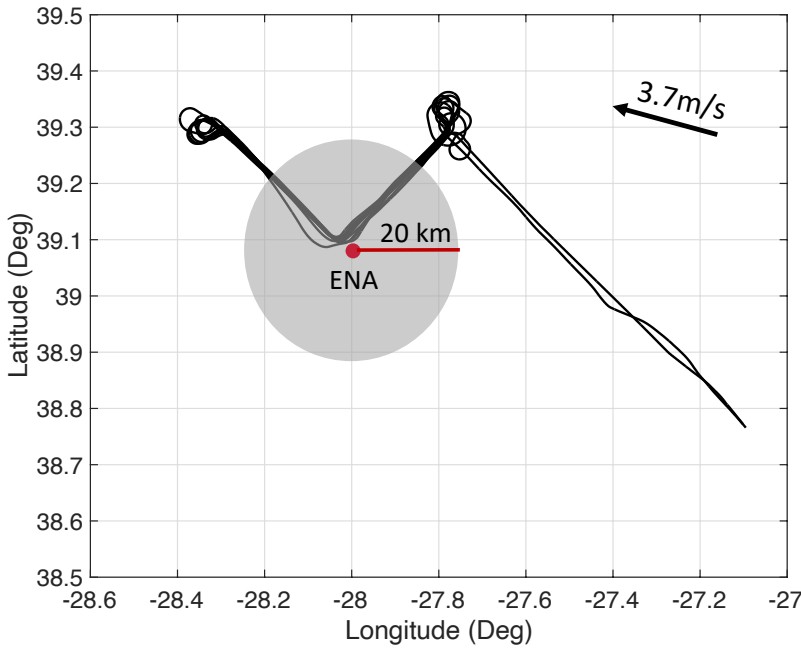

Figure 4: Aircraft track (black line) during the operational period on June 30, 2017. Shaded circle
indicates the area within 20km around ENA site. The arrow in the upper right corner indicates
mean wind direction and wind velocity in cloud layer during the observational period.


















Table 2: Selected stratocumulus days in ACE-ENA, MAGIC and MARCUS campaigns.

| ARM site | Selected Days |
|---|---|
| ENA | 20170603, 20170604, 20170605, 20170616, 20170617, 20170627,20170628, 20170630, 20170701,20170702, 20160703, 20170706, 20170707, 20170709, 20170713,20170714, 20170715, 20170718, 20170719 |
| MAGIC | 20121016, 20121020, 20121030, 20121105, 20130526, 20130604,20130605, 20130708, 20130709, 20130710, 20130717, 20130720, 20130721,20130722, 20130729, 20130730, 20130731, 20130804 |
| MARCUS | 20180109, 20180110 ,20180228, 20180301, 20180322, 20180323 |















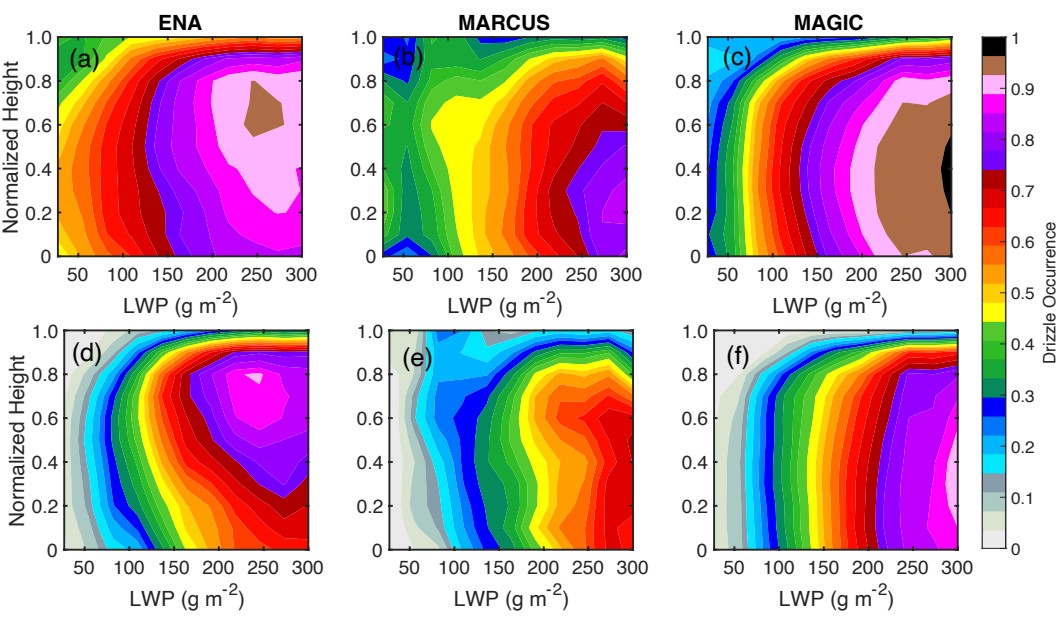


Figure 5: Vertical distribution of drizzle occurrence categorized by LWP based on MLA under
ENA (a), MARCUS (b) and MAGIC (c) observational campaigns. (d), (e) and (f) are same as (a),
(b), (c) except the drizzle is detected by dBZA.








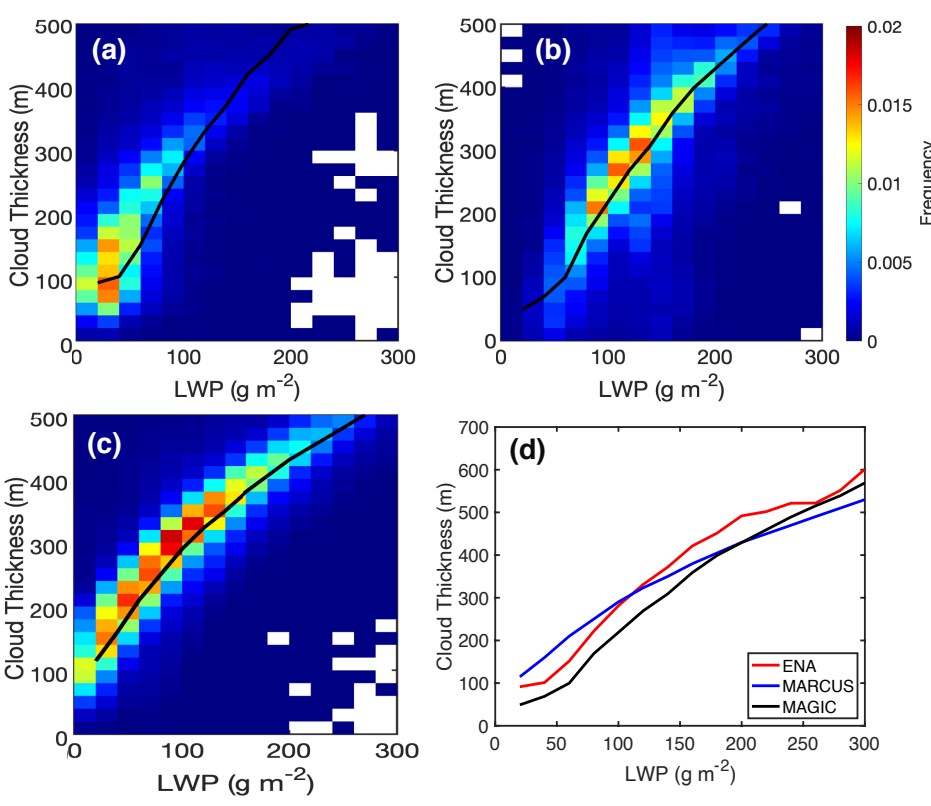

Figure 6: Joint histogram of cloud thickness and LWP at three campaigns: (a) ENA, (b) MARCUS
and (c) MAGIC. The black line indicates the mean cloud thickness in each LWP category. For
comparison, the relationship between mean cloud thickness and LWP at three campaigns (black
line in (a),(b),(c) ) are shown in (d).










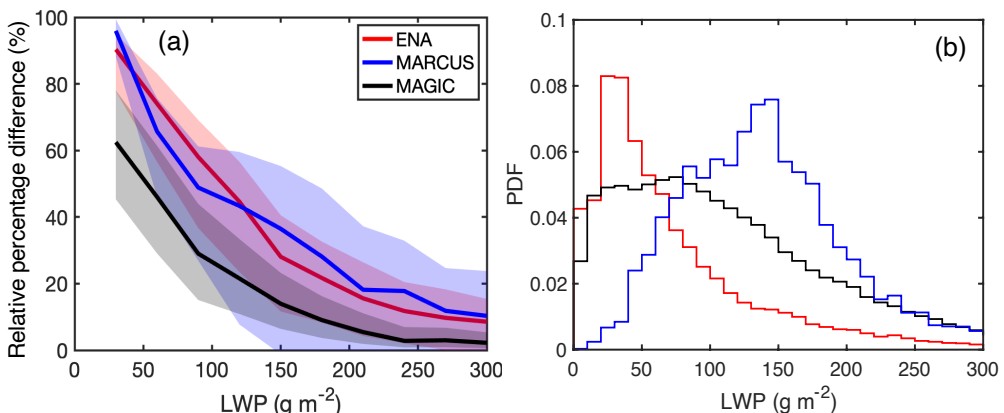


Figure 7: (a) Relative percentage difference of drizzle detection between the dBZA (dBZ > -17) and MLA as a function of LWP at ARM observational campaigns: ENA (red line), MARCUS (blue line) and MAGIC (black line). The shading area indicates same results but with different reflectivity threshold being used: the upper boundary is for the dBZ > -15 and the lower boundary is for dBZ > -20. (b) Histogram of the LWP distribution collected at three campaigns: ENA (red line), MARCUS (blue line) and MAGIC (black line).

696

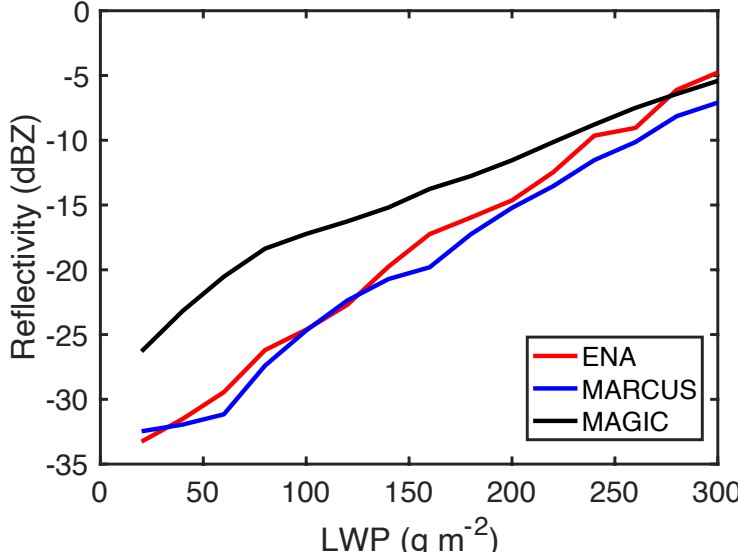

Figure 8: Mean KAZR reflectivity of the hydrometeor signal as a function of LWP at three campaigns: ENA (red line), MARCUS (blue line) and MAGIC (black line).

699