# Peer review of "New insights on the prevalence of drizzle in marine stratocumulus"

_Atmospheric Chemistry and Physics, 2021_

## Referee Comment (RC1)

I really enjoyed reading the manuscript, that is presenting an innovative methodology to classify drizzle presence in the clouds using a machine learning algorithm. The presented results are robust and potentially open new future studies and analysis. I find the paper well written and I only have some small minor comments regarding the exposition of some scientific concepts. Despite the completeness of the exposition, I could find a couple of open questions regarding the machine learning algorithm and the physical interpretation of the classification, that I would like to pose to the authors, for possibly improving the manuscript further. Please find below the detailed list of my comments.
Best regards

**Technical comments:**

line 55: what is autoconversion? At this line the term is introduced without any previous definition. To make the reading easier, you might first define what is the autoconversion process. I mean, it does not exist in nature as a process… it is model representation of the droplet growth… maybe some reflections on this could help the reader to understand better the problem you are tackling.

Line 57-58: cloud organizational structure… do you mean cloud spatial organization? And the boundary layer system… is it the boundary layer structure?

Line 70: Van Zanten , not VanZanten, there's a space missing, I guess.

Line 99: To maybe integrate on what you wrote: The threshold chosen on the skewness is based on the estimations of the skewness variability in non-drizzling conditions derived in Acquistapace et al., 2017 (https://doi.org/10.5194/amt-10-1783-2017). In that work, in section 3.1.1 we explained how to obtain that threshold: "Standard deviations of the skewness time series in the nondrizzling cloud using 2s integration time and spectral resolutions of 256, 512 and 1024 range between 0:389 and 0:369 with a mean value over the three cases of 0:379." Maybe this helps to make the threshold choice clearer.

Line 170: I don't understand how the sigma-t is derived. Which retrieval do you apply?  and why you do not use a set of sigma t values instead of just one? I am asking because turbulence can strongly alter the spectra shape and it can be used to test your algorithm using values in the tails of your distribution.

**Comment relative to figure 3.C:**

From your classification, drizzle seems to occur anywhere in the cloud. Are you able to interpret this evidence in a physical way? How is this evidence for example correlated with turbulence in the cloud? Is there any statistical correlation between more turbulent bins and machine learning classified pixels? A possible approach to test this would be to derive EDR using Borque et al, 2016 approach for example.

**General questions:**

1) Why did you pick the combination of parameter Gamma=50 C=1, giving 0.95 for precision and 0.85 for recall? Many other combinations of the parameters give the same outcome or even better ones (for example, from the table, I could spot gamma = 1, C = 1000). How do the other choices affect the results? Can you justify more why you pick your choice? It would help to better understand the role of gamma and C to have a plot showing the variability of the black line with respect to the choices of gamma, and C, for maybe some sets.

2) how does the reflectivity profile of the clouds whose radar bins fall in the region between -30 and -20 dBz (that you highlight in the nice figure 2) look like? In our past work, we used the cloud adiabaticity (slope of the gradient of Ze in the profile) to identify clouds in which the embryonic drizzle onset was starting but no skewness signature was already evident. I would be interested to know if there's any correspondence. This point comes back to the physical interpretation of what's happening when drizzle develops, and in which conditions. The idea was that when drops grow with diffusion of water vapor they follow an adiabatic profile, and when other interactions take over, the profile ceases to be adiabatic.

---

## Author Comment (AC1)

I really enjoyed reading the manuscript, that is presenting an innovative methodology to classify drizzle presence in the clouds using a machine learning algorithm. The presented results are robust and potentially open new future studies and analysis. I find the paper well written and I only have some small minor comments regarding the exposition of some scientific concepts. Despite the completeness of the exposition, I could find a couple of open questions regarding the machine learning algorithm and the physical interpretation of the classification, that I would like to pose to the authors, for possibly improving the manuscript further. Please find below the detailed list of my comments.

Dear Dr. Acquistapace,

We appreciate the time and the efforts you have dedicated to providing valuable feedback on our manuscript. The manuscript has been modified and much improved based on those valuable comments. A point-by-point response (text in blue) to the comments can be found below.

Technical comments:

line 55: what is autoconversion? At this line the term is introduced without any previous definition. To make the reading easier, you might first define what is the autoconversion process. I mean, it does not exist in nature as a process... it is model representation of the droplet growth... maybe some reflections on this could help the reader to understand better the problem you are tackling.

**Response**: We appreciate the review's suggestions. We have added more discussions of the autoconversion process in the introduction:

"….One important issue is the representation of the transition from cloud water to rainwater, which is parametrized by the autoconversion process via different schemes…"

Line 57-58: cloud organizational structure... do you mean cloud spatial organization? And the boundary layer system... is it the boundary layer structure?

Response: We thank the reviewer for pointing out the inappropriate usage of these terminologies. Modifications have been made based on the suggestions:

"…Drizzle is frequently observed in the warm cloud system and can modulate the cloud spatial organization and the boundary layer structure in several ways:…"

Line 70: Van Zanten , not VanZanten, there's a space missing, I guess.

**Response:** We believe that the citation format here is consistent with the previous literatures.

Line 99: To maybe integrate on what you wrote: The threshold chosen on the skewness is based on the estimations of the skewness variability in non-drizzling conditions derived in Acquistapace et al., 2017 (https://doi.org/10.5194/amt-10-1783-2017). In that work, in section 3.1.1 we explained how to obtain that threshold: "Standard deviations of the skewness time series in the nondrizzling cloud using 2s integration time and spectral resolutions of 256, 512 and 1024 range

between 0:389 and 0:369 with a mean value over the three cases of 0:379." Maybe this helps to make the threshold choice clearer.

**Response**: We thank the reviewer for this valuable supplementation. We modified the description of the threshold selection as follow:

"…The capability of using skewness to detect early drizzle development stages was demonstrated in Acquistapace et al. (2019), where a skewness threshold as 0.379 was estimated from the standard deviation of the Doppler skewness time series with carefully selected nondrizzling clouds (Acquistapace et al., 2017)…"

Line 170: I don't understand how the sigma-t is derived. Which retrieval do you apply? and why you do not use a set of sigma t values instead of just one? I am asking because turbulence can strongly alter the spectra shape and it can be used to test your algorithm using values in the tails of your distribution.

**Response**: For the vertical pointing radar, the observed spectrum width is a measure of the Doppler spectrum broadening which is contributed by three factors: turbulence ($\sigma_t$), microphysics (i.e., the falling velocity difference among hydrometers with different size) and the wind shear effects (usually is negligible compared to other two terms) (Borque et al., 2016). In our study, we assume that Doppler spectrum broadening is mainly contributed by the turbulence factor in the non-drizzling or weakly drizzling clouds, and thus the observed second-moment of the Doppler spectrum, i.e. spectrum width, can be used to indicate the turbulence broadening factor ($\sigma_t$).

It is correct that in the presence of the strong turbulence, Doppler spectrum tends to smooth out and leads to a smaller skewness. In our study, we are using one fixed turbulence broadening factors as this value represents the general turbulence environment for the clouds of interest (i.e. stratocumulus clouds).

We modified the $\sigma_t$ retrieval description according to the reviewer's suggestion in the text:

"…For the vertical pointing radar, the observed spectrum width is a measure of the Doppler spectrum broadening which is mainly contributed by three factors: turbulence ($\sigma_t$), microphysics (i.e., the falling velocity difference among hydrometers with different size) and the wind shear effects (usually is negligible compared to other two terms) (Borque et al., 2016). In our study, we assume that Doppler spectral broadening is mainly contributed by the turbulence factor in non-drizzling (or weakly drizzling) clouds, and thus the observed second-moment of the Doppler spectrum, i.e. spectrum width, can be directly used to indicate the turbulence broadening factor ($\sigma_t$). The mean value of the KAZR-observed spectrum width collected from the ACE-ENA IOP1 is estimated as 0.2 m/s (Fig. S1). Thus, $\sigma_t$ is selected as 0.2m/s for the Doppler spectrum simulator to represent the typical turbulence environment for the stratocumulus clouds of interest…"

Comment relative to figure 3.C:

From your classification, drizzle seems to occur anywhere in the cloud. Are you able to interpret this evidence in a physical way? How is this evidence for example correlated with turbulence in the cloud? Is there any statistical correlation between more turbulent bins and machine learning classified pixels? A possible approach to test this would be to derive EDR using Borque et al, 2016 approach for example.

**Response**: We appreciate this valuable suggestion. The effect of the dynamics on the drizzle formation is not the focus of this study, more related work will be shown in our future research. Here we are pleasure to provide a simple analysis on this topic as posed by the reviewer.

We use the case as shown in Fig. 3 as an example. Cloud as observed in Fig. 3 is very weak and more than 99% of the reflectivity signals are lower than -20 dBZ, thus the microphysics has negligible effect on the observed spectrum width, which can be used to indicate the turbulence intensity (according to our previous assumption). Figure R1 (a) shows the PDF of spectrum width and (b) indicates the relationship between the detected drizzle occurrence from the ML algorithm and the observed spectrum width (e.g. turbulence intensity). In Figure R1 (b), drizzle occurrence increases monotonously as spectrum width increase, indicating that drizzle is more favorable to be formed in a strong turbulence environment.

[Figure]

Figure R1: Left: PDF of Doppler spectrum width of the drizzle signals for case 20170630. Right: Drizzle occurrence as a function of spectrum with.

We also added the following discussion in the conclusion to indicate the potential application of this study on the future research.

"…The ubiquitous of drizzle in the MBL clouds calls for investigations on the drizzle formation mechanism. It is known that the growth of liquid droplets by diffusion is not efficient with radius larger than 20 $\mu m$, thus other mechanisms that favors drizzle formation greatly contribute the drizzle existence. The presented results provide observational evidence to verify the drizzle formation theories…"

General questions:

1) Why did you pick the combination of parameter Gamma=50 C=1, giving 0.95 for precision and 0.85 for recall? Many other combinations of the parameters give the same outcome or even better ones (for example, from the table, I could spot gamma = 1, C = 1000). How do the other choices affect the results? Can you justify more why you pick your choice? It would help to better understand the role of gamma and C to have a plot showing the variability of the black line with respect to the choices of gamma, and C, for maybe some sets.

**Response**: We thank the reviewer for this valuable suggestion. We added Fig. S3-Fig. S8 in the supplement using the same ML algorithm but with different C and $\Gamma$ as listed in Table 1. Generally, these two parameters (C and $\Gamma$) control the complexity of the resolved boundary shape with large C and $\Gamma$ usually leads to overfitting. We also modified the main text to clarify the chosen of the ML parameters:

"…Besides using the metrics as recall/precision, the shape of the resolved boundary is also examined visually to avoid the ML algorithm being overfitted. As shown in Figs. S3~ S8. Parameter with large C and $\Gamma$ leads to better classification outcome but will cause overfitting issues…"

2) how does the reflectivity profile of the clouds whose radar bins fall in the region between -30 and -20 dBz (that you highlight in the nice figure 2) look like? In our past work, we used the cloud adiabaticity (slope of the gradient of Ze in the profile) to identify clouds in which the embryonic drizzle onset was starting but no skewness signature was already evident. I would be interested to know if there's any correspondence. This point comes back to the physical interpretation of what's happening when drizzle develops, and in which conditions. The idea was that when drops grow with diffusion of water vapor they follow an adiabatic profile, and when other interactions take over, the profile ceases to be adiabatic.

**Response**: We want to thank the review's for posting this interesting topic. Fig. R2(a) shows the drizzle signals detected by the ML algorithm with reflectivity between -30 and -20 dBZ.; Fig. R2(b) shows the vertical gradient of the reflectivity profile. It can be seen that for the "weak drizzle" signals, the reflectivity gradient is positive in clouds except near cloud top where strong entrainment process happens. This result indicates that the presence of the drizzle particles can also generate a positive reflectivity gradient profile. Given the relatively large radar observational volume (30m vertical and 2s temporal resolution), it is highly likely that once the drizzle embryos are generated, they will mix with cloud particles and increase the reflectivity in the given radar volume.

[Figure]

Figure R2: (a) Reflectivity of the Drizzle signals as identified in Fig. 3, noted that only drizzle with reflectivity between -30 and -20 dBZ are shown. (b): Vertical gradient of the reflectivity profile of the "weak drizzle" shown in Fig. R2(a).

Acquistapace, C., Kneifel, S., Löhnert, U., Kollias, P., Maahn, M., and Bauer-Pfundstein, M.: Optimizing observations of drizzle onset with millimeter-wavelength radars, Atmospheric Measurement Techniques, 10, 1783-1802, 2017.

Acquistapace, C., Löhnert, U., Maahn, M., and Kollias, P.: A new criterion to improve operational drizzle detection with ground-based remote sensing, Journal of Atmospheric and Oceanic Technology, 36, 781-801, 2019.

Borque, P., Luke, E., and Kollias, P.: On the unified estimation of turbulence eddy dissipation rate using Doppler cloud radars and lidars, Journal of Geophysical Research: Atmospheres, 121, 5972-5989, 2016.

---

## Author Comment (AC2)

General comments:

This a very important study that will have a significant impact on the field. The article is well structured and written. I have no major comments, just a few minor suggestions. I would like to propose that the article is accepted after the minor changes.

We appreciate the reviewer's acknowledgement on our work. The manuscript has been modified and much improved based on those valuable comments. A point-by-point response (text in blue) to the reviewer's comments can be found below.

My main suggestion is to use standard verification metrics when presenting the performance of the method, i.e. In Fig. 2. I would suggest that you show probability of detection, false alarm rate and critical success index as functions of Z, instead of CDF.

Response: We thank the review's suggestions. In Fig.2 we want to highlight the significant difference between the ML and the conventional drizzle detection method, thus we think it is more intuitive and appropriate to show the CDF of the detected drizzle signal as a function of reflectivity. We followed the review's suggestions by adding the standard verification metrics for the two-drizzle detection methods as shown in Fig S9 in the supplement. We also modified the text as follows:

"…A more detailed performance comparisons of the two drizzle detection methods are shown in Fig. S9, where the results are similar with Fig.2, the rise of the false detection rate for the ML-based method for reflectivity lower than -20dBZ is due to the exists of the extremely weak drizzle signals as will be discussed later…"

Minor comments:

Line 157: "...cloud/drizzle datasets is trained by a machine learning algorithm..."

Do you mean a machine learning algorithm is trained by the datasets?

**Response**: We thank the reviewer for pointing out this oversight, the modification has been made:

"…Finally, a machine learning algorithm is trained by the collection of well-defined cloud/drizzle datasets to resolve the drizzle identification function…."

Line 170: "...turbulence broadening is set as 0.2 m/s which is obtained from local observations..."

You use spectra width for cases where Z is less than -20 dBZ, do you do any other data filtering? As you show later, spectra shape might be modified by auto conversion even for such low reflectivity values.

**Response:** We didn't perform other filtering process as we believe reflectivity < -20 dBZ is an adequate constrain to isolate the effect of microphysics on Doppler spectrum width. For the vertical pointing radar, the observed spectral width is a measure of the Doppler spectrum broadening which is mainly contributed by three factors: turbulence ($\sigma_t$), microphysics (i.e., the falling velocity difference among hydrometers with different size) and the wind shear effects (usually is negligible compared to other two terms). In our study, we assume that for the non-drizzling (or weakly drizzling) clouds, Doppler spectrum broadening is mainly contributed by the turbulence factor, thus the observed second-moment of the Doppler spectrum, i.e. spectral width, can be used to estimate the turbulence broadening factor ($\sigma_t$).